# Arabidopsis Root Development Regulation by the Endogenous Folate Precursor, Para-Aminobenzoic Acid, via Modulation of the Root Cell Cycle

**DOI:** 10.3390/plants12244076

**Published:** 2023-12-05

**Authors:** Hanna Lasok, Hugues Nziengui, Philip Kochersperger, Franck Anicet Ditengou

**Affiliations:** 1Malopolska Centre of Biotechnology, Jagiellonian University, Gronostajowa 7A, 30-387 Kraków, Poland; hanialasok@gmail.com; 2Faculty of Biology, Institute of Biology II, Albert Ludwigs University Freiburg, 79104 Freiburg, Germany; 3Department of Biology, Faculty of Sciences, Science and Technology University of Masuku, Franceville P.O. Box 913, Gabon; hugues.nzienguiikapi@univ-masuku.org; 4Lighthouse Core Facility, Medical Center University of Freiburg, Albert Ludwigs University Freiburg, 79106 Freiburg, Germany; 5Bio Imaging Core Light Microscopy (BiMiC), Institute for Disease Modelling and Targeted Medicine (IMITATE), Medical Center University of Freiburg, Albert Ludwigs University Freiburg, 79106 Freiburg, Germany

**Keywords:** *Arabidopsis thaliana*, PABA, root growth, folates, cell cycle

## Abstract

The continuous growth of roots depends on their ability to maintain a balanced ratio between cell production and cell differentiation at the tip. This process is regulated by the hormonal balance of cytokinin and auxin. However, other important regulators, such as plant folates, also play a regulatory role. In this study, we investigated the impact of the folate precursor para-aminobenzoic acid (PABA) on root development. Using pharmacological, genetic, and imaging approaches, we show that the growth of *Arabidopsis thaliana* roots is repressed by either supplementing the growth medium with PABA or overexpressing the PABA synthesis gene *GAT-ADCS*. This is associated with a smaller root meristem consisting of fewer cells. Conversely, reducing the levels of free root endogenous PABA results in longer roots with extended meristems. We provide evidence that PABA represses Arabidopsis root growth in a folate-independent manner and likely acts through two mechanisms: (i) the G2/M transition of cell division in the root apical meristem and (ii) promoting premature cell differentiation in the transition zone. These data collectively suggest that PABA plays a role in Arabidopsis root growth at the intersection between cell division and cell differentiation.

## 1. Introduction

The uptake of water and nutrients, as well as the overall performance of a plant, are determined by its root system architecture (RSA) [1]. The latter describes the organisation of primary and lateral roots, adventitious roots, and root hairs. The primary root is established from the root apical meristem (RAM) during embryo development, whereas lateral roots are initiated post-embryonically from pericycle lateral root founder cells of the primary root [2].

The entire primary root results from the activity of the stem cell niche in the RAM. The root apex consists of five distinct functional/developmental zones based on their cellular activities: the meristematic zone (MZ), containing fast proliferative cells; the root transition zone (TZ), composed of cells containing small vacuoles and exhibiting slow cell growth in both length and width; the fast elongating zone (EZ), which contains fast elongating cells but no change in width; the growth terminating zone (GTZ), where cells progressively slow down their elongation to reach their mature lengths and exhibit root hair tip growth; and the differentiation zone (DZ), containing fully differentiated cells [3,4,5,6].

The integrity of the RAM is crucial for continuous root growth and relies on a balance between the rate of cell differentiation and the rate of cells undergoing mitosis. This balance is established through the antagonistic effects of auxin, which promotes cell division, and cytokinins, which promote cell differentiation [7,8]. The position of the TZ, at the boundary between cell proliferation and cell differentiation, is tightly regulated. This regulation requires cytokinins to inhibit auxin signalling through the cytokinin response regulator ARR1-regulated expression of the auxin-conjugating *GH3.17* and the negative regulator of auxin signalling *SHY2* genes [7,9,10,11,12].

Maintenance of the RAM depends on proper functioning of the cell division cycle, which consists of four stages: an increase in cell size (Gap 1), DNA synthesis (Synthesis or the S stage), cell growth and preparation for cell division (Gap 2), and cell division (Mitosis or the M stage) [13]. The G0 stage is a non-proliferative state during which the cell enters a quiescent phase. Heterodimeric Ser/Thr protein kinases, comprising the catalytic subunit cyclin-dependent kinase (CDK) and the activating subunit cyclin (CYC), play a crucial role in driving cells through the successive stages (G1/S/G2/M) of the cell cycle [14]. In Arabidopsis, these cyclins play a regulatory role in root growth. The double-mutant *cycb1;1 cycb1;2* exhibits a 50% reduction in root growth compared to the wild type [15]. Cyclins are synthesised and degraded in a cyclic manner during the cell cycle, and their concentrations are the highest during the specific phase of the cell cycle in which they are acting. When the level of a specific cyclin decreases, the associated CDKs become less active, which leads to the cell entering the next stage of the cell cycle [13]. CDK–CYC complexes are important targets for both environmental and internal cellular signals. CDKs’ cell division-promoting activity is suppressed by various pathways that act in parallel [16].

Several marker lines have been developed in Arabidopsis to visualise the spatiotemporal progression of the cell cycle. The use of a truncated version of the DNA replication origin licensing factor *CDT1a*, driven by the S phase-specific promoter of a histone 3.1-type gene, has been reported as a reliable marker for the S + G2 phases [17]. Otero et al. [18] demonstrated that the histone H3.1 protein is maintained during the M and G1 phases in frequently dividing cells of the root meristem. However, it is expelled during the G2 phase in cells undergoing their final cell cycle before differentiation in the root TZ. When used in combination with CYCB1;1, which is specific to the G2/M transition, it is possible to visualise both the S to G2 and G2 to M cell cycle stages [17,18]. More recently, Desvoyes et al. (2020) developed the first Plant Cell Cycle Indicator (PlaCCI) marker line. This marker line expresses *CDT1a-CFP*, the histone protein H3.1 (*HTR13-mCherry*), and *CYCB1;1-YFP*, driven by their endogenous promoters. This development convincingly enables the identification of cell cycle stages in the Arabidopsis root tip [19].

Alongside phytohormones, studies have also revealed that the folate derivative 5-formyl-tetrahydrofolate is an essential regulator of root meristem maintenance [20,21]. Folates are tripartite molecules composed of para-aminobenzoic acid (PABA), pterine, and one or several glutamate moieties [22]. They are essential elements in the biosynthesis of lipids, chlorophyll, and lignin, as well as the regulation of gene and protein expression [23,24]. These components, which are essential parts of the human diet due to their health benefits, are synthesized de novo only in plants, fungi, and bacteria. The first step in the folate biosynthesis pathway takes place in the cytosol, where pterine biosynthesis occurs, leading to the formation of 6-hydroxymethyldihydropterin (HDMDHP). This compound is then phosphorylated in the mitochondria to form 6-hydroxymethyldihydropterin pyrophosphate (HMDHP-P2) (reviewed by [23]).

Folate precursor PABA is synthesised in plastids from chorismate and glutamine through a two-step reaction. This reaction is successively mediated by AMINODEOXYCHORISMATE LYASE (ADCL) and bifunctional GLUTAMINE AMIDOTRANSFERASE–AMINODEOXYCHORISMATE SYNTHASE (GAT-ADCS) [25,26]. More than 80% of PABA is esterified by URIDINE DIPHOSPHATE (UDP)-GLUCOSYLTRANSFERASE (UGT75B1) to form PABA-glucose, which is then stored in the vacuole [23,27,28]. The remaining portion is utilized in the biosynthesis of folates within the mitochondria, where PABA combines with HMDHP-P2 to produce dihydropteroate (DHP). Afterward, a glutamate residue attaches to the carboxyl group of PABA in a reaction catalysed by DIHYDROFOLATE REDUCTASE (DHFR) to produce tetrahydrofolate (THF). THF is then linked to additional glutamate molecules in mitochondria, plastids, or vacuoles [29,30].

It has long been believed that PABA activity is closely associated with the folate biosynthesis pathway. However, in previous work, we demonstrated that PABA promotes asymmetric root growth during root gravitropism independently from folates. This suggests that endogenous PABA can act as a signalling molecule to modulate Arabidopsis root growth [31]. In the present study, we further examine the role of PABA in root growth and development. We demonstrate that artificially elevating PABA levels through the external addition of PABA to the growth medium or overexpression of the PABA synthesis gene *GAT-ADCS* significantly impedes root growth, and this is correlated with a smaller root meristem size due to fewer cells. On the contrary, reducing free PABA levels in the roots by conjugating PABA to glucose results in longer roots with extended meristems. In addition, we provide evidence that PABA activity affects root meristem size by deregulating the progression of root cell cycles through two mechanisms: (i) inhibiting the G2/M transition and (ii) promoting the premature differentiation of cells in the transition zone. Taken together, these data support the role of the folate precursor PABA in Arabidopsis root growth at the transition from cell division to differentiation.

## 2. Results

### 2.1. PABA Reduces the Number of Cells in the Root Meristem

We previously reported that the primary root of Arabidopsis is inhibited by PABA treatment in a dose-dependent manner [31]. To investigate the impact of PABA on root elongation, wild-type Arabidopsis (Col-0) seedlings were grown in the presence of PABA concentrations ranging from 0 to 400 µM (Figure 1A). The elongation of the primary root was measured over time, starting when the root radicle broke through the seed coat. Studying root elongation over time revealed that exogenously applied PABA repressed root growth in a time- and concentration-dependent manner (Figure 1B). Root growth inhibition was delayed in the presence of 50 µM PABA and only occurred on day 6, while concentrations higher than 50 µM significantly repressed root growth starting from day 3 (Figure 1B). These data suggest that root inhibition begins very early, possibly when the emerging radicle enters into contact with the PABA medium.

Since the elongation of the primary root depends on cell production in the root apical meristem (RAM), we quantified the size of the root meristematic zone (MZ) at day 6 by measuring and summing up the length of individual cortex cells in continuity, starting from the cortex cell initial to the first elongated cell (dashed line in Figure 1C). As expected, the inhibition of root elongation by PABA correlated well with the shortening of the MZ, with the highest reduction in meristem size (~33%) observed at 400 µM. The application of 200 µM, 100 µM, and 50 µM PABA resulted in a 25%, 8%, and 5% reduction in meristem size, respectively (Figure 1D). In addition, PABA also had a negative impact on the number of root cortex cells. Non-treated roots consisted of a meristem of 43 ± 5 cortex cells (Figure 1E); this number is reduced to 40 ± 5 cells in the presence of both 50 µM and 100 µM PABA (Figure 1E). The number of cells dramatically dropped by 12 and 16 cells at 200 µM and 400 µM PABA, respectively (Figure 1E). Because root growth was significantly reduced (almost stopped) from 200 µM, we deemed this concentration optimal for the rest of the study, as using higher concentrations could potentially lead cytotoxic effects.

### 2.2. PABA Regulation of Root Growth Is Independent of Folates

To get insights into whether the inhibitory effect of PABA on root growth is related to its role as a folate precursor [26], Arabidopsis plants were grown for 7 days in the presence of PABA or 5-formyltetrahydrofolic acid (5-FTHF), a stable folate derivative that occurs naturally and can be easily converted into metabolically active folates once it is taken up by cells [26,32]. PABA, but not 5-FTHF, repressed root growth and significantly affected both RAM size and the number of meristematic cells (Appendix A). This suggests that PABA and 5-FTHF have different effects on root growth. Interestingly, when 1 mM 5-FTHF was applied, it promoted root growth, suggesting that 5-FTHF acts as a positive regulator of root development.

### 2.3. PABA Negatively Affects the G2/M Transition of Root Meristematic Cells

To check whether PABA acts on cell proliferation, we used Arabidopsis transgenic lines carrying *pCYCB1;2::CYCB1;2-GUS* and *pCYCB1;2::CYCB1;2-GFP* constructs, respectively, where the *CYCB1;2* promoter and coding region, including the cyclin destruction box (CDB), is fused in frame to GUS [33] or GFP [34]. This enabled the visualisation of the G2-M cell cycle transition and the degradation of the resulting CDB-GUS or CDB-GFP protein at the end of mitosis [34]. Overall, both the intensity of the GUS signal of CDB-GUS roots, as determined by quantifying the mean grey values of the GUS signal [35], and the GFP signal intensity in GFP-expressing roots were significantly lower in the presence of PABA (Figure 2A–D). Taken together, these data suggest that PABA negatively impacts the G2/M transition of meristem cells.

To investigate the impact of PABA on other phases of the cell cycle, we utilised the Plant Cell Cycle Indicator (PlaCCI) marker line [19]. PlaCCI and wild type (WT) were equally sensitive to 200 µM PABA, and no significant difference in root length was measured (Appendix A). Next, in line with Desvoyes et al. (2020) [19], confocal microscopy images showed that the root tip of non-treated PlaCCI roots could be divided into three main regions (Figure 3): (i) region 1, containing highly proliferative cells characterized by the expression of all markers, *CDT1a-CFP* (G1 phase), *HTR13-mCherry* (G1-S-G2-M phases), and *CYCB1;1-YFP* (G2/M phase); (ii) region 2, occupied by cells in G1-phase, expressing *CDT1a-CFP* and *HTR13-mCherry*, but not *CYCB1;1-YFP*; (iii) and finally, region 3, devoid of *CYCB1;1-YFP* expression, containing cells in the G and S phases undergoing endocycle (orange dashed line in Figure 3A,B and Appendix A). In PABA-treated roots, region 1, albeit being smaller than in control roots, and region 2 could be easily recognized (Figure 3C,D and Appendix A). However, in line with a shorter meristem (magenta dashed line), the distal boundary of region 3, as indicated by the cells expressing *CDT1a-CFP* (white arrow in Figure 3A–D and Appendix A), was much closer to the root tip. This suggests that meristem cells undergo earlier differentiation. Measuring the distance between the quiescent centre (QC) and the first cell expressing *CDT1a-CFP* confirmed this statement (Figure 3E).

Finally, to confirm whether cells had indeed differentiated earlier, we quantified the distance between the tip of the root and the first bulging root hair. We found that the root hairs bulged significantly closer to the root tip on PABA, as shown in Figure 3F.

### 2.4. Endogenous PABA Is a Key Component in the Regulation of Arabidopsis Root Growth

PABA is synthesised through a two-step reaction involving the plastidial bifunctional enzyme glutamine amidotransferase–aminodeoxychorismate synthase (GAT-ADCS) [25,26]. We previously reported that the *GAT-ADCS* gene is expressed in the QC, root cap cells, and root stem cells [31]. On the other hand, the dihydropteroate synthase (*DHPS*) gene, which is involved in PABA incorporation into the folate pathway, is expressed in columella cell initials and root cap cells [31]. This strongly suggests that PABA is indeed produced in the stem cell niche and meristematic zone. To investigate the significance of endogenous PABA on root growth, we created transgenic lines that overexpress the *GAT-ADCS* gene. Among the 19 lines analysed (Appendix A), we isolated the 35S::ADCS-12 with high levels of *GAT*-*ADCS* mRNA (Appendix A), which accumulates twice as much PABA as the WT Col-0 (Appendix A). Overexpression of *GAT-ADCS* resulted in a reduction in both shoot and root growth (Figure 4A), suggesting that PABA content might be critical for shoot development in addition to its role in the root. The root length, the size of the root meristem, and the number of cells in the meristem were significantly reduced by 30–40% in this line (Figure 4B–D). Similar results, but not as pronounced, were obtained using a mutant lacking the UDP-GLYCOSYL TRANSFERASE75B1 (UGT75B1; [28]), the enzyme that catalyses the glycosylation of PABA, resulting in plants over-accumulating free PABA [28]. Compared to the wild type (WT), the *Landsberg erecta* (L*er*) *ugt75b1* knock-out mutant developed primary roots that were 15% shorter, with a reduction in meristem size by 5% and a decrease in the number of meristematic cells by 10% (Figure 4E–H).

Conversely, to artificially reduce the amount of free PABA, we utilised a stable conditional line that overexpresses *UGT75B1* (Lex::UGT75B; [31]). This line, when compared to the wild type (WT), accumulates PABA glucosylates (PABA-Glc) upon β-estradiol induction [31]. In the absence of β-estradiol, the roots of Lex::UGT75B plants appeared slightly longer, with larger meristems containing more cells than WT plants (Figure 5A–D). This trend can be explained by the reported leakiness of the β-estradiol inducible system [31,36]. At 0.1 µM, β-estradiol had no impact on the growth of the WT root, but it stimulated a 14% increase in the elongation of the Lex::UGT75B primary root (Figure 5A,B). This stimulation was only 7% greater when compared to non-induced Lex::UGT75B control roots (Figure 5A,B). The promotion of root growth was accompanied by an increase in meristem size and the number of meristematic cells (Figure 5C,D). However, increasing β-estradiol concentration to 0.5 µM significantly impacted the growth of both WT and Lex::UGT75B roots, and the differences reported above became less pronounced (Figure 5A–D). The primary root of Lex::UGT75B was only 7% longer than that of β-estradiol-treated WT plants, and the size of the meristem and the number of meristematic cells were about 10% higher compared to WT (Figure 5B–D). These data suggest that a slight decrease in free PABA levels leads to an increase in meristematic cell number and meristem size, promoting root growth. Conversely, increasing free PABA levels would reduce cell cycle activity, resulting in a decrease in the number of meristematic cells and limiting root growth. Maybe the rate of free PABA conversion to PABA-Glc in the presence of 0.5 µM β-estradiol leads to a deficiency of free PABA, which is also necessary for proper cell division, root meristem maintenance, and root development and maintenance.

## 3. Discussion

The makeup of the plant relies on its ability to convert both biotic and abiotic signals into developmental responses. Therefore, the sustained growth of the root, which results from a balance between the production of new cells in the RAM and their progressive differentiation in the TZ, is a fundamental asset for plant adaptability to various ecological niches. However, the molecular and physiological mechanisms that sustain the continuous growth of the root are not fully understood. In this study, we explored the role of the folate precursor PABA in root growth and development using pharmacological, genetic, and imaging approaches.

### 3.1. PABA Represses G2/M Transition and Promotes the Endocycle

Because PABA-treated roots are shorter (this study; [31,37]) and display a smaller meristem, which consists of fewer cells compared to untreated roots, we hypothesised that a defect in cell division alters the development of these roots. Plants contain a large number of cyclins, the vast majority of which are uncharacterized. During the G2 phase of the cell cycle, B-type cyclins are synthesised and reach their peak levels during prometaphase. This is the stage when the chromosomes condense and the spindle apparatus forms, preparing the cell for mitosis. We found that the number of cells expressing the B-type cyclin CYC B1;1, which is crucial for the G2/M cell cycle transition, was dramatically reduced in PABA-treated roots. As a consequence, the number of cells undergoing the G2/M transition is indeed significantly reduced in the presence of PABA. CYCB1;1 is synthesised during the G2 phase, peaks during the prometaphase, and disappears at early anaphase [38]. Hence, by affecting CYCB1;1 expression, PABA prevents meristem cells from fully completing their mitotic cycle, thereby causing them to lose their proliferation potential.

In untreated roots, the transition between cell division and cell elongation is progressive as cells enter the elongation zone. They express the DNA replication origin licensing factor CDT1a and the G2 marker H3.1, but not the G2/M marker CYCB1;1 (present study; [17,19]), indicating their G1 status and that they are undergoing the endocycle [19]. This is in line with the shortening of the MZ and the precocious development of root hairs. The fact that this transition occurs much closer to the root tip in PABA-treated plants suggests that PABA promotes the shift in meristematic cells from the mitotic cell cycle to the endocycle, an alternate cell cycle during which cells replicate their genetic material but do not split [39]. Thus, by repressing CYCB1;1, PABA inhibits cells from completing mitosis and cytokinesis, while still allowing for normal chromosome replication [40]. Indeed, we never observed apoptotic cells, and the fact that PABA only slows down the growth, even when applied at high concentrations, also supports the hypothesis that cells are alive but divide at a low rate. However, it cannot be excluded that when occurring, high concentrations of free PABA might be hindered by increased levels of PABA glucosylation by the UGT75B enzyme. How PABA acts on CYCB1;1 is still an open question. However, we showed in a previous study that PABA activity on root gravitropism requires ethylene biosynthesis [31]. This is particularly interesting since ethylene, known to repress the growth of the Arabidopsis root, was reported to act through a mechanism involving the reduction in the CYCB1;1 protein level [41]. This highlights the possibility that PABA may modulate root growth in an ethylene-dependent post-translational regulatory mechanism controlling CYCB1;1 stability [41].

### 3.2. PABA and Auxin Display Distinct Activities on Root Growth

The importance of plant hormone crosstalk in maintaining the continuous growth of the root has been clearly demonstrated [42,43,44]. For instance, the action of cytokinins through the activation of SHY2/IAA3 (SHY2), the negative regulator of PIN auxin transporters genes, by primary cytokinin response transcription factor ARR1 in the root transition zone, can reduce auxin distribution and thereby promote cell differentiation [10]. The gaseous hormone ethylene also can repress root elongation by two distinct mechanisms. A significant portion of the inhibitory ethylene effect on root growth was suggested to be mediated by the local stimulation of the auxin response in the meristem and EZ epidermis [42,45]. However, the existence of yet-to-be-described auxin-independent mechanisms was also evidenced [45].

Interestingly, using theoretical calculations and crystallographic analyses, it was reported that PABA in its anionic form (PABA^−^) would bind the active site of the auxin receptor TIR1, similarly to the naturally occurring auxin, indole acetic acid (IAA) [37]. This led to the conclusion that PABA might act as a potential auxin-like plant growth regulator [37,46]. IAA indeed inhibits root growth even at very low concentrations (nanomolar range) [47,48,49], whereas PABA inhibition of root growth is visible from 50 μM (this study; [31,37]). This concentration-dependent activity may account from the fact that IAA exhibits stronger binding to TIR1 than PABA [31]. However, contrary to the previously mentioned inhibitory effect of PABA on cell division, it was observed that elevated concentrations of auxin stimulate the root cell cycle and impede the transition to the endocycle through the activation of CDKs and mitotic cyclin production in unison with cytokinin [10,50,51,52]. Consistent with this, disrupting auxin response in root cell files results in the repression of cell cycle genes, including *CYCB1;1* [52,53]. Thus, although both compounds inhibit root growth, one (auxin) positively influences cell division and negatively endo-reduplication, whereas the other (PABA) inhibits cell cycle-G2/M-transition and promotes cell differentiation.

Further, we provide evidence that root endogenous PABA levels are important for the balance proliferation/differentiation of cells at the root tip. Artificially increasing or decreasing endogenous PABA levels, respectively, reduces or promotes meristem size. PABA acts in a folate-independent manner since PABA activity on meristem size maintenance is distinct from that of tetrahydrofolate. PABA is synthesised in QC and lateral root cap cells, and is probably redistributed at the root tip through diffusion. Indeed, as a weak acid, PABA might enter cells in its uncharged form [31].

### 3.3. Conclusions and Future Prospects

In this study, we present a mechanism by which the folate precursor PABA modulates root development by regulating the G2/M transition of the cell cycle. The activity of PABA on the cell cycle appears to be similar to that of ethylene, suggesting that PABA may act, at least partially, upstream of the ethylene pathway. It is interesting to note that cells enter the endocycle when they pass through a region with an auxin minimum, which represents the boundary between proliferating and differentiating cells [6]. We do not yet know with certainty whether PABA interacts with auxin in this region. Further studies are needed to determine whether PABA also affects cell elongation, which appears to be the primary target of auxin [42]. However, our previous study demonstrated that PABA’s effect on root gravitropic response relies on the auxin biosynthesis *TAA1* gene expression in the transition zone, likely through an ethylene-dependent mechanism [31]. Therefore, an increase in PABA-promoted auxin levels could, in theory, result in a PABA-induced auxin-dependent inhibition of cell growth in the root region. Altogether, these observations support the idea that PABA might regulate root growth through two mechanisms involving both auxin and ethylene signalling. Specifically, PABA would act at the point where cell proliferation and cell differentiation intersect by repressing cell growth and inhibiting cell division.

## 4. Materials and Methods

### 4.1. Plant Material and Growth Conditions

Apart from *ugt75b* (GT6017) in the *Landsberg erecta* background, all genetic backgrounds were from the Arabidopsis (*Arabidopsis thaliana*) ecotype Col-0. pCYCB1;2:: CYCB1;2-GUS [33], pCYCB1;2:: CYCB1;2-GFP [34], Lex::UGT75b [31], and PlaCCl seeds [19] were kindly provided by Prof. C. Gutierrez (Centro de Biologia Molecular Severo Ochoa, CSIC-UAM, Cantoblanco, Madrid, Spain). Seeds were surface-sterilised using a solution of 5% (*w*/*v*) calcium hypochlorite and 0.02% (*v*/*v*) Triton X-100 in 80% (*v*/*v*) ethanol. Then, they were rinsed twice with 80% (*v*/*v*) ethanol and once with 100% (*v*/*v*) ethanol. For all experiments, seedlings were grown vertically in Petri dishes tilted at an angle of 60° from the vertical axis. The seedlings were placed on a solid 1.3% (*w*/*v*) agar Arabidopsis medium, which was prepared by adding 0.5 × basal salt Murashige and a Skoog medium supplemented with 1% (*w*/*v*) sucrose and 5 mM MES (pH 5.8). The growth conditions were maintained in a climatic cabinet at 21 °C, under long-day conditions (16 h of light and 8 h of dark), with a humidity level of 70%.

### 4.2. Construction of Plasmids

Plasmids were constructed according to the protocol described by Sambrook et al. [54]. Gateway^®^ cloning procedures (Invitrogen, Waltham, MA, USA) were used, and *E. coli* TOPO10 cells (Invitrogen) were used as the host for the constructed plasmids. GAT-ADCS cDNA was obtained from RIKEN (Tokyo, Japan), amplified using cADCS_F (CTGTCGTTATTCGGAATGATGAGT) and cADCS_R (GAAAGCAGCTCTTACATTCCCAC) oligonucleotides and cloned into the pDONR207 vector. Subsequently, it was cloned into the pMDC32 vector to generate the 35S::GAT-ADCS construct.

### 4.3. Plant Transformation

Arabidopsis plants were transformed using the agrobacterium strain GV3101 (pMP90) according to the standard floral dip method [55]. Transformants were selected on 15 μg/mL Hygromycin B.

### 4.4. Plant Scanning and Microscopy

Seedlings were imaged using a flatbed CanonScan 9950F scanner (Canon Inc., Öta, Japan) while they were growing on a plastic Petri dish. Histological detection of GUS activity and plant preparation for microscopy were performed according to Ditengou et al. [56]. For light microscopy, samples were observed using a Zeiss Axiovert 200M MOT device (Carl Zeiss MicroImaging, Jena, Germany) to capture high-magnification images. Low-magnification views were obtained using a Zeiss Stemi SV11 Apo stereomicroscope (Carl Zeiss MicroImaging) and observed under differential interference contrast optics. Plants expressing fluorescent proteins were stained with 5 µg/mL FM4-64, a lipophilic probe that binds to cell plasma membranes. The stained plants were then analysed using Nikon’s AZ-C1 Macro Laser Confocal Microscope and the Zeiss LSM 980 laser scanning microscope. To simultaneously monitor DAPI, YFP, and mCherry fluorescence, we employed multitracking in-frame mode, and the emission was separated using the META spectral analyser online unmixing feature. Images were extracted and analysed using Zen software version 3.4 (Carl Zeiss MicroImaging, Jena, Germany) and Imaris 9.8.0 (Oxford Instruments, Abingdon, UK).

### 4.5. Quantification of Root Meristem Size and Number of Meristematic Cells

To measure the size of the root meristem and the number of meristematic cells, we collected roots from Arabidopsis seedlings of different genotypes that were grown for 6 days with or without PABA. The collected roots were then stained with 5 µM FM4-64 for 5 min. After a quick rinse, the roots were scanned using a Nikon C1 confocal microscope (NIKON, Tokyo, Japan). The size of the meristem and the number of cortex cells were calculated following the method described by Takasuka et al. [57].

### 4.6. Quantification of GFP Signals

GFP quantifications were performed by selecting a meristem region that included the quiescent centre (QC) and stem cells up to the first elongated cortex cell of equally sized root tips. Fluorescence intensity was measured as mean pixel intensities in 8-bit images containing only the GFP channel using ImageJ software (https://imagej.net/ij/ij/) [58].

### 4.7. Quantification of GUS Signals

GUS quantifications were performed by selecting a meristem region that included the quiescent centre (QC) and stem cells up to the first elongated cortex cell. Signal intensity was measured using ImageJ software, as described by Beziat et al. [35].

### 4.8. Quantification of Free PABA

Free and total PABA determinations were essentially performed according to the method described by Camara et al. [26].

### 4.9. Figures and Statistical Analyses

All figures were created using Inkscape software version 1.2.2 (Free Software Foundation, Inc., Boston, MA, USA). Box plots were generated using the BoxPlotR web-tool [59] and GraphPad Prism V.5.0. The latter program was also used for all statistical analyses.

### 4.10. Accession Numbers

The sequence data from this article can be found in the GenBank/EMBL database and the Arabidopsis Genome Initiative database under the following accession number: At1g05560 (UGT75B1).

## Figures and Tables

**Figure 1 plants-12-04076-f001:**
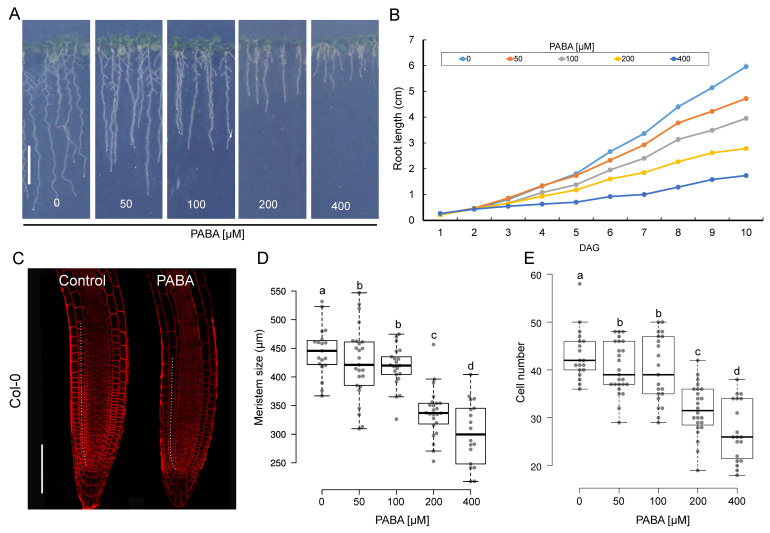
The impact of PABA on Arabidopsis primary root growth. (**A**) Root size of 6-day-old plants grown in 0 (control), 50, 100, 200, and 400 μM PABA-supplemented media. (**B**) Primary root elongation over time (*n* = 45). Day after germination (DAG). (**C**) Representative root tips of 6-day-old plants stained with the plasma membrane marker FM4-64 (red), grown in the absence (control) or presence of 200 μM PABA. The white dashed lines depict the limits of the meristematic zone. Bar, 100 µm. (**D**,**E**) Size of the meristem and number of cortex cells in the meristem. Data are shown as the means ± standard error (*n* ≥ 20 roots). The letters (a–d) indicate that the meristem size or cortex cell number differs from the control, as determined by a one-way ANOVA with the Tukey multiple testing corrections (*p* < 0.05).

**Figure 2 plants-12-04076-f002:**
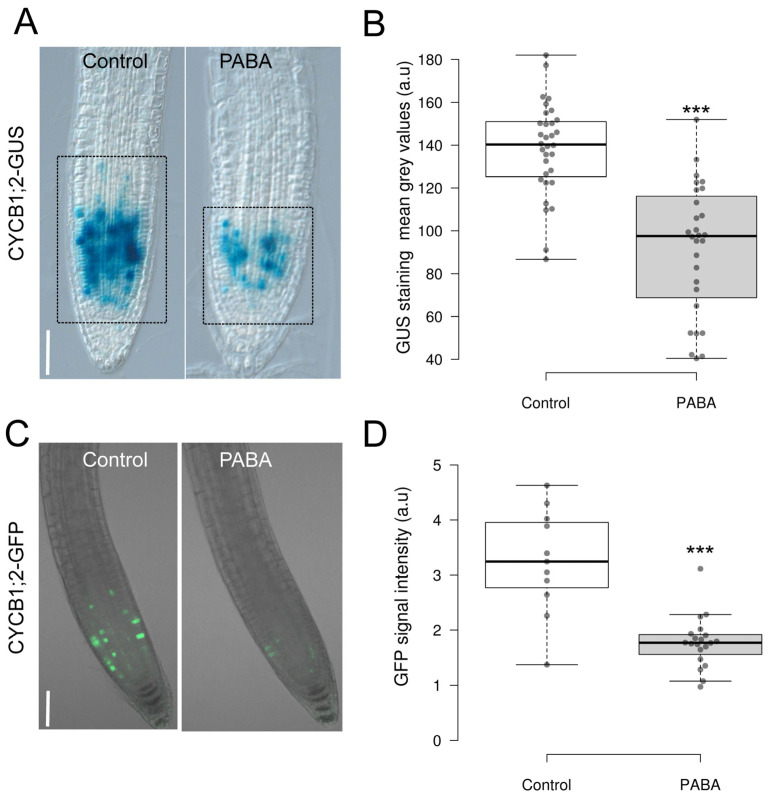
Expression pattern of CYCB1 at the root tip. (**A**) Visualisation of GUS staining of 6-day-old seedlings expressing the *pCYCB1;2::CYCB1;2-GUS-CDB* construct grown on control and 200 μM PABA media. (**B**) Quantification of the GUS signal from plants shown in (**A**). (**C**) Visualisation of the *pCYCB1;2::CYCB1;2-GFP* expression in dividing cells (green) in 6-day-old seedlings grown in the absence (control) and presence of 200 μM PABA. (**D**) Quantification of the GFP signal from plants shown in (**C**). Data are shown as the means ± standard error (*n* = 28 for (**B**) and ≥11 for (**D**)). Asterisks indicate statistical significance based on Student’s *t* test (*** *p* < 0.001). a.u., arbitrary units. Scale, 100 µm.

**Figure 3 plants-12-04076-f003:**
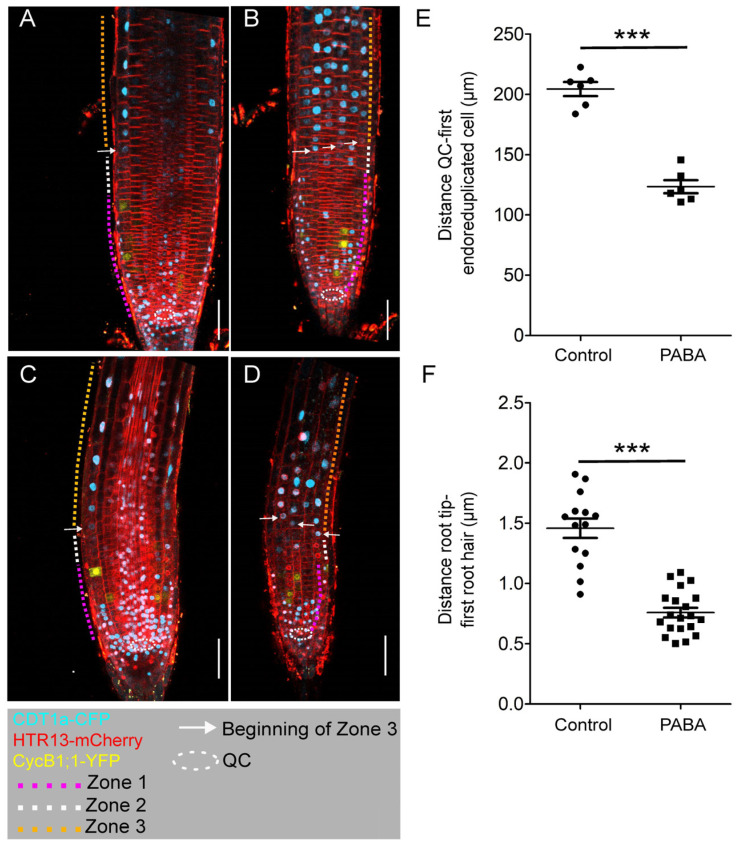
Use of the PlaCCI reporter line to visualise the root cell cycle state. Confocal images displaying the root tips of 6-day-old PlaCCI plants expressing *pCDT1a::CDT1a-eCFP*, *pHTR13::HTR13-mCherry*, and *pCYCB1;1::NCYCB1;1-YFP* constructs and grown in the absence (control) or presence of 200 μM PABA. (**A**,**B**) Control root. (**A**) Middle section of the control root. (**B**) Root epidermis and cortex. (**C**,**D**) PABA-treated roots. (**C**) Middle section of the root. (**D**) Root epidermis and cortex. (**E**) Distance between the QC and the first endo-reduplicated cell. The asterisks indicate statistical significance based on Student’s *t* test (*n* = 6; *** *p* < 0.001). (**F**) Distance between the root apex and the first bulging root hair (*n* = 20; *** *p* < 0.001). Dashed circle line depicts the position of the quiescent centre (QC). Dashed lines mark the limits of the root zones: magenta (MZ, zone 1), white (TZ, zone 2), and orange (zone 3, differentiation zone). The white arrow indicates the beginning of the differentiation zone. Bar, 20 μm.

**Figure 4 plants-12-04076-f004:**
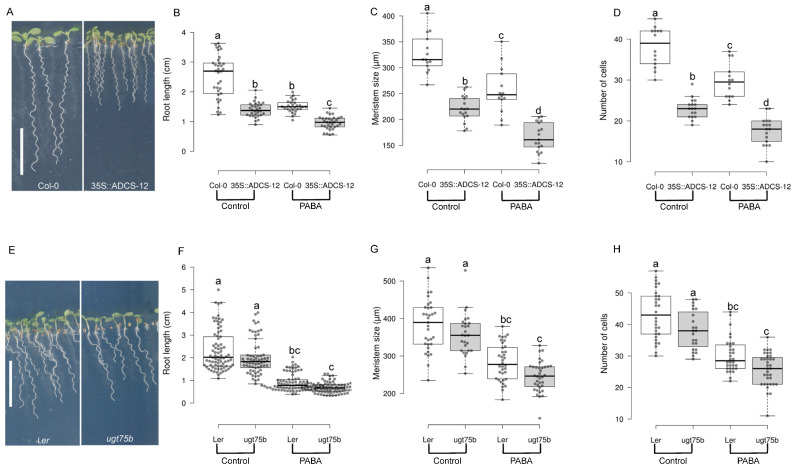
Endogenous PABA regulates root growth. Analysis of root development of Arabidopsis wild type, 35S::ADCS-12 transgenic, and *ugt75b1* knock-out lines. (**A**–**D**) Phenotype analysis of WT (Col 0) versus 35S::ADCS-12. (**A**) Col 0 (WT, left panel) and 35::ADCS-12 plants grown in the absence of PABA. (**B**) Quantifications of the root length of plants shown in (**A**). (**C**) Size of the meristem. (**D**) Number of meristematic cells. (**E**–**H**) Phenotype analysis of WT (L*er*) versus *ugt75b1* loss-of-function mutant. (**E**) L*er* (left panel) and *ugt75b1*. (**F**) Quantifications of the root length of plants shown in (**E**). (**G**) Size of the meristem. (**H**) Number of meristematic cells. White box plots indicate WT Col-0 (**B**,**C**,**D**) and L*er* (**F**,**G**,**H**) plants, while grey box plots represent 35S::ADCS-12 (**B**,**C**,**D**) and *ugt75b1* (**F**,**G**,**H**) plants, respectively. Data are shown as the means ± se (*n* = 30 for (**B**,**F**); *n* = 16 for (**C**,**D**,**G**,**H**)). The letters (a–d) indicate that the root length, meristem size or cortex cell number differs from the control, as determined by a one-way ANOVA with the Tukey multiple testing corrections (*p* < 0.05). Bar, 1 cm.

**Figure 5 plants-12-04076-f005:**
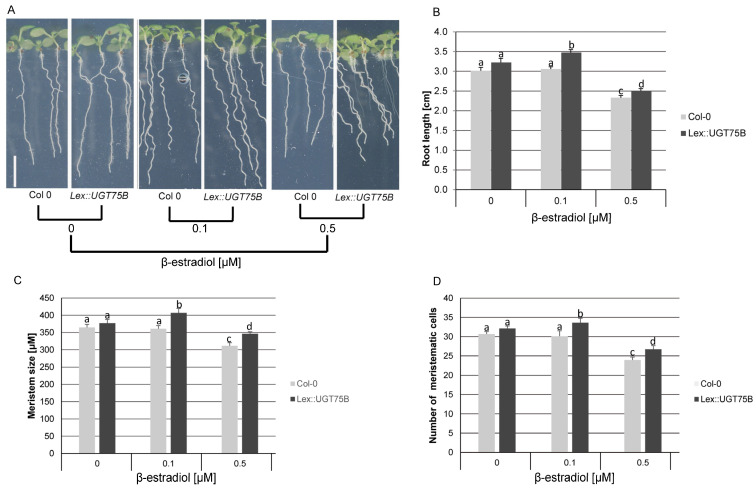
Effect of reduced free endogenous PABA levels on root development. (**A**) The phenotype of Arabidopsis WT (Col-0) and Lex::UGT75B plants grown in the absence (0) or in the presence of β-estradiol (0.1 and 0.5 μM). (**B**–**D**) Quantification of root length, meristem size, and number of meristematic cells, respectively. Data are shown as the means ± se (*n* = 35 roots). The letters (a–d) indicate that the root length, meristem size or cortex cell number differs from the control, as determined by a one-way ANOVA with the Tukey multiple testing corrections (*p* < 0.05). Bar, 1 cm.

## Data Availability

Data are available upon request from the corresponding author.

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
