# Peer review of "Arabidopsis Root Development Regulation by the Endogenous Folate Precursor, Para-Aminobenzoic Acid, via Modulation of the Root Cell Cycle"

_plants, 2023, doi:10.3390/plants12244076_

Round 1

Reviewer 1 Report

Comments and Suggestions for Authors

Dear authors, thanks a lot for submitting this manuscript. I have made suggestions in the pdf document (visible as comments in acrobat). I do not think the experiments shown are done in sufficient replicates or have high enough n numbers. Moreover, it is not clear how the novelty of these conclusions need a separate small manuscript. I therefor am only giving it a low score on scientific soundness and interest to readers. Based on that I would not recommend the current manuscript for publication as such, but part of a more extensive study the data are valuable.

Comments on the Quality of English Language

Dear authors,

I have made formating and English corrections in the pdf document shared. These can be visualized in Acrobat, menu VIEW-TOOLS-Comment.

Author Response

Dear Reviewers,

Here we resubmit our paper entitled "Regulation of Arabidopsis Root Development by the Endogenous Folate Precursor, Para-aminobenzoic Acid, through Modulation of the Root Cell Cycle." We would like to thank you for your valuable comments, which we believe improved the manuscript. We hope the article now meets all the requirements, allowing it to be published in Plants.

Sincerely,

Franck Ditengou

Reviewer 2 Report

Comments and Suggestions for Authors

The manuscript entitled “Arabidopsis root development is regulated by the endogenous 2 folate precursor, para-aminobenzoic acid, through modulation 3 of the root cell cycle” reports a fundamental research done in the model plant species, Arabidopsis thaliana, based on pharmacological, physiological and molecular approaches. The results are clear, the manuscript it is well written and easy to understand for the readers. However, in my opinion, the inclusion of some aspects, details and minor corrections, presented below, can improve the quality of the manuscript.

Abstract

Please write PABA in full and indicate the species name in italics.

Results and Discussion sections

Lines 132/133: In understood the idea but please clarify the sentence.

Lines 146/147: Explain better in the results (and in the discussion section) why the concentration of PABA used in the further analyses was only 200 µM. The reduction of meristem size and number of meristematic cells were similar between the 200 µM and 400 µM treatments. In addition, why the PABA concentrations lower than 200 µM were not used in the further analyses?

Line 153: Please indicate standard error (s.e.)

Figure 1-B: Please indicate in the caption of Figure 1 what means DAG.

Line 172: WT?

Line 184: QC?

Line 193: DHPS? gene

Please write these and other acronyms in full when they appear for the first time in the manuscript because their full names are presented in the material and methods section that it is at the end of the manuscript.

Lines 180/81: Please revise the sentence.

Please revise the caption of Figure 4, it is not clear and some parts are repeated.

Lines 278-282: Please analyse the article from Janeczko, A. Estrogens and Androgens in Plants: The Last 20 Years of Studies. Plants 2021, 10, 2783. https://doi.org/10.3390/ plants10122783, where this author showed that a high concentration of estrogens can have a harmful effect on plant growth, morphology and development (including root growth and development), to present another alternative explanation for the result achieved with 0.5 µM of β-estradiol and to improve its discussion. This higher concentration can be causing cytotoxicity fur to oxidative stress explaining the reduced cell division rate and the DNA damage may cause cell cycle arresting in the G2 phase.

Lines 153-154; 290-291: Please remove results from the caption of figures. Put the PË‚0.05 value at the end of the sentence that refers that different lowercase letters or asterisks indicate significant statistical differences among treatments or refer that these tests were performed for a significance level of 95%.

Lines 359 and 427: Please indicate the references for the Inkscape software and also for the ImageJ software.

Lines 439 and 443: Please indicate details about the measurement of the intensity of the fluorescence signals of GUS and GFP with the ImageJ software.

Lines 456-459: I was not able to analyse the supplementary material, it was not provided and the link ii is not functioning.

Supplementary Materials: The following supporting information can be downloaded at: 456 www.mdpi.com/xxx/s1, Figure S1: Comparison of the impact of PABA and 5-FTHF on the root 457 growth; Figure S2: Use of the PlaCCI reporter line to visualize the root cell cycle state. Figure S3: 458 Characterization of ADCS overexpressing lines.”

Conclusions

The authors developed a fundamental research that enriched the knowledge of the root cell cycle and growth regulation using the plant model species Arabidopsis. Therefore, how the present findings can contribute for the crops productivity given the high importance of the root system for the uptake of water and nutrients under the context of the climate changes? 

Materials and methods

Please indicate the total number of seeds allowed to germinate; the number of seeds and the number of seedlings used per experiment. A variable number of roots are presented through the results section for each experiment. Since no germination data was provided, the number of achieved roots was reduced to enable the realisation of all experiments in all concentrations. This could be explained in materials and methods or results. Otherwise, indicate the number of roots analysed per experiment in the M&M section.

Author Response

Dear reviewer 2,

Here we resubmit our paper entitled "Regulation of Arabidopsis Root Development by the Endogenous Folate Precursor, Para-aminobenzoic Acid, through Modulation of the Root Cell Cycle." We would like to thank you for your valuable comments, which we believe improved the manuscript. We hope the article now meets all the requirements, allowing it to be published in Plants.

Sincerely,

Franck Ditengou

Reviewer 3 Report

Comments and Suggestions for Authors

The manuscript of Lasok et al. examines the role of the folate precursor PABA in root growth and development. Exogenous application of PABA prevented root growth by G2/M transition inhibition and early differentiation of meristem cells. Likewise, overexpression of the GAT-ADCS resulted in PABA overaccumulation and caused root growth reduction. In consistency, mutation of UGT75B1 that also led to accumulation of free PABA, had an impact on root growth. On the contrary, induction of UGT75B1, stimulating PABA conjugation and decreasing the free levels of PABA, led to marginal but significant stimulation of root growth. Overall, root endogenous PABA levels are important for the balance proliferation/differentiation of cells at the root tip.  

The manuscript is well written and the experiments support the rationale and the conclusions. There are only few minor comments that could improve the manuscript.

#01: The introduction looks long. It would be beneficial for the article to transfer several lines to the discussion session.

#02, Figure 3:

(i) The findings and the biological significance of the fluorescent protein markers of Figure 3 should be elaborated within lines 217-218, 229-230 and 208-212, respectively.

(ii) The legend explains dashed lines with white (TZ, zone 2) and orange (zone 3, differentiation zone) color, but these are not clearly evident in Fig. 3A-D.

#03, Figure S2C legend: What about the panels on the right?

Author Response

Dear reviewer 3,
Here we resubmit our paper entitled "Regulation of Arabidopsis Root Development by the Endogenous Folate Precursor, Para-aminobenzoic Acid, through Modulation of the Root Cell Cycle." We would like to thank you for your valuable comments, which we believe improved the manuscript. We hope the article now meets all the requirements, allowing it to be published in Plants.

Sincerely,
Franck Ditengou
